# Is educational attainment associated with the onset and outcomes of low back pain? a systematic review and meta-analysis

Aliyu Lawan[1], Alex Aubertin[2], Jane Mical[1], Joanne Hum[3], Michelle L. Graf[4], Peter Marley[1], Zachary Bolton[1], David M. Walton[1]*

1 Faculty of Health Sciences, School of Physical Therapy, Western University, London, Ontario, Canada,
2 School of Health Sciences, Nursing and Emergency Services, Cambrian College, Sudbury, Ontario, Canada, 3 Palliative Care, Fraser Health Authority, New Westminster, British Columbia, Canada,
4 Department of Physical Therapy, University of British Columbia, Vancouver, British Columbia, Canada

* dwalton5@uwo.ca

**Data Availability Statement:** The data underlying the results presented in the study are available within the manuscript.

## Abstract

### Background

Low back pain (LBP) is the leading global cause of years lived with disability. Of the biopsychosocial domains of health, social determinants of LBP remain under-researched. Socioeconomic status (SES) may be associated with the onset of new LBP or outcomes of acute LBP, with educational attainment (EA) being a key component of SES. The association between EA and LBP has yet to be the subject of a dedicated review and meta-analysis.

### Purpose

To review evidence of the association between EA and a) onset or b) outcomes of acute and subacute LBP in the adult general population and to conduct statistical pooling of data where possible.

### Methods

An electronic search was conducted in MEDLINE, Embase, CINAHL, and ProQuest from inception to 2$^{nd}$ November 2023 including reference lists to identify relevant prospective studies. Risk of bias (RoB) was assessed using the Quality in Prognostic Studies (QUIPS) tool. Where adequate data were available, estimates were pooled using a random-effects meta-analysis. Overall evidence for each outcome was graded using an adapted GRADE.

### Results

After screening 8498 studies, 29 were included in the review. Study confounding and attrition were common biases. Data from 19 studies were statistically pooled to explore EA as a predictor of new LBP onset or as prognostic for outcomes of acute or subacute LBP. Pooled results showed no association between EA and the onset of new LBP (OR: 0.927, 95%CI: 0.747 to 1.150; I$^2$ = 0%). For predicting outcomes of acute LBP, compared to those with no more than secondary-level education, post-secondary education or higher was associated with better outcomes of pain (OR: 0.538, 95%CI: 0.432 to 0.671; I$^2$ = 35%) or disability (OR:

**Funding:** The author(s) received no specific funding for this work.

**Competing interests:** The authors have declared that no competing interests exist.

0.565, 95%CI: 0.420 to 0.759; $I^2$ = 44%). High heterogeneity ($I^2$>80%) prevented meaningful pooling of estimates for subacute LBP outcomes.

## Conclusion

We found no consistent evidence that lower EA increases the risk of LBP onset. Lower EA shows a consistent association with worse LBP outcomes measured at least 3 months later after acute onset with inconclusive findings in subacute LBP. Causation cannot be supported owing to study designs. High-quality research is needed on potential mechanisms to explain these effects.

## Introduction

Low back pain (LBP) is a leading global cause of years lived with disability [1]. In North America, chronic LBP is amongst the top ten reasons for seeking medical attention [2] with a prevalence of 18–23% in adults Canadians [3]. While many LBP cases resolve within the first three months, it has been estimated that as many as 60% to 80% progresses to chronicity or recurrence within one year including loss of productivity in 40% [4,5]. The acute to chronic transition of LBP is a complex process with multiple mechanisms likely influencing the pathway including biological, psychological, and social [6–8]. Prevention of new LBP or prevention of the acute-to-chronic transition stand to have a major impact on global health burden [2]. Existing guidelines recommend early identification of psychosocial factors that could prevent or enhance recovery from LBP [9].

While the biological and psychological sciences have provided considerable evidence to explain onset of and recovery from LBP, much less attention has been paid to the social influences. Social determinants of health (SDOH) are increasingly recognized as potent influences on the genesis of several health states [10], with some prior authors indicating that neighborhood characteristics may have at least as large an influence on the experience of chronic diseases as do personal genetics [11]. While SDOHs represent a large and complicated field of research, there are some social variables unique to the person experiencing pain that are worthy of dedicated inquiry. One such variable is educational attainment (EA), defined as the highest level of education completed by a person. As a prognostic variable, EA represents a blend of person-level (e.g., literacy) and society-level (e.g., access) influences and could potentially hold value as a variable through which intervention strategies could be tailored. EA holds value for research on SDOHs as "years of education" is one of few such variables that can be readily quantified.

There is some evidence that chronic pain is more prevalent amongst people with low EA [12] and lower EA may predict the acute-to-chronic transition [13]. However, there are limited studies that focus solely on the social predictors of LBP specifically [6]. EA has been included in some prior systematic reviews in LBP [14,15] though differences in case definitions, variable definitions, or study design have precluded clear findings. Even rarer are reviews or evidence syntheses on the association between EA and the onset of new LBP in population-level cohort studies that start with pain-free participants [16]. If lower EA is a risk for new onset LBP or for poor recovery following onset of acute LBP, mechanisms could then be explored and if causation is supported EA could be integrated into either public health prevention strategies or tailored treatment planning to prevent the acute-to-chronic transition.

The purpose of this systematic review was to qualitatively and/or quantitatively synthesize published estimates on the risk and prognostic value of EA on the onset or outcomes of LBP.

## Methods

### Design

This review was designed and reported according to the Preferred Reporting Items for Systematic Reviews and Meta-Analyses (PRISMA) framework [17]. The review was limited to observational prospective cohort or population-level studies (not clinical trials) of patients aged $\geq 18$ years with either no LBP at inception (followed to determine onset), or acute (<8 weeks) and/or sub-acute (8–12 weeks) LBP. We focused on 'non-specific' LBP and therefore excluded LBP related to underlying systemic medical (such as cancer, infection, or cauda equina syndrome), vertebral compression fracture or osteoporosis, inflammatory conditions (e.g., ankylosing spondylitis) or neurological conditions (e.g., stroke). Beyond that, we accepted the case definitions of LBP as reported by the authors of the primary sources.

### Search strategy

The search strategy (Appendix A) was developed with a research librarian using MeSH terms specific to MEDLINE which was adapted for other databases. No specific restrictions on publication date were set. The search strategies were applied to MEDLINE (OVID), The Cumulative Index of Nursing and Allied Health Literature (CINAHL), EMBASE, and ProQuest from inception to 30th March 2023 and updated to 2nd November 2023 corresponding to the dates of the respective searches without restriction to language of publication. A grey literature search of unpublished studies was conducted in Researchsquare. Hand searches of reference lists of all included articles were conducted to identify additional primary sources.

### Study selection

Yield from each database were imported into Covidence systematic review software (Veritas Health Innovation, Melbourne Australia) and screened by two independent reviewers against the inclusion criteria, with disagreements being resolved by a third reviewer. Titles and abstracts were screened to remove irrelevant sources, followed by full-text screening against the inclusion/exclusion criteria. Kappa was calculated as an indicator of agreement between raters. The reasons for exclusion are included in Fig 1.

### Risk of bias in individual studies

The Quality in Prognostic Studies (QUIPS) tool was used to assess RoB of all included studies. QUIPS consists of six category-domains of potential biases: i) study participation, ii) attrition, iii) prognostic factor measurement, iv) outcome measurement, v) confounding, and vi) statistical analysis/reporting. All included studies were assessed by 2 independent reviewers. We used a worst-score approach, where each paper was assigned a RoB based on the worst (highest risk) rating of any of the 6 categories [18] classed as low, moderate, or high risk of bias (RoB). RoB agreement was calculated through Cohen's kappa with disagreements resolved through discussion with a third experienced reviewer.

### Data extraction

Data were extracted using a study-specific extraction table that included key study descriptors, sample characteristics, operationalizations of EA and LBP, outcomes assessed, and relevant findings. Educational attainment was extracted with as much detail as reported in the publication. Where possible, the minimum data extracted were related to a 12-year cut-point for EA as representing the threshold between secondary (up to year 12) and post-secondary (beyond year 12) EA in most countries. Where data were not presented with adequate detail, EA was

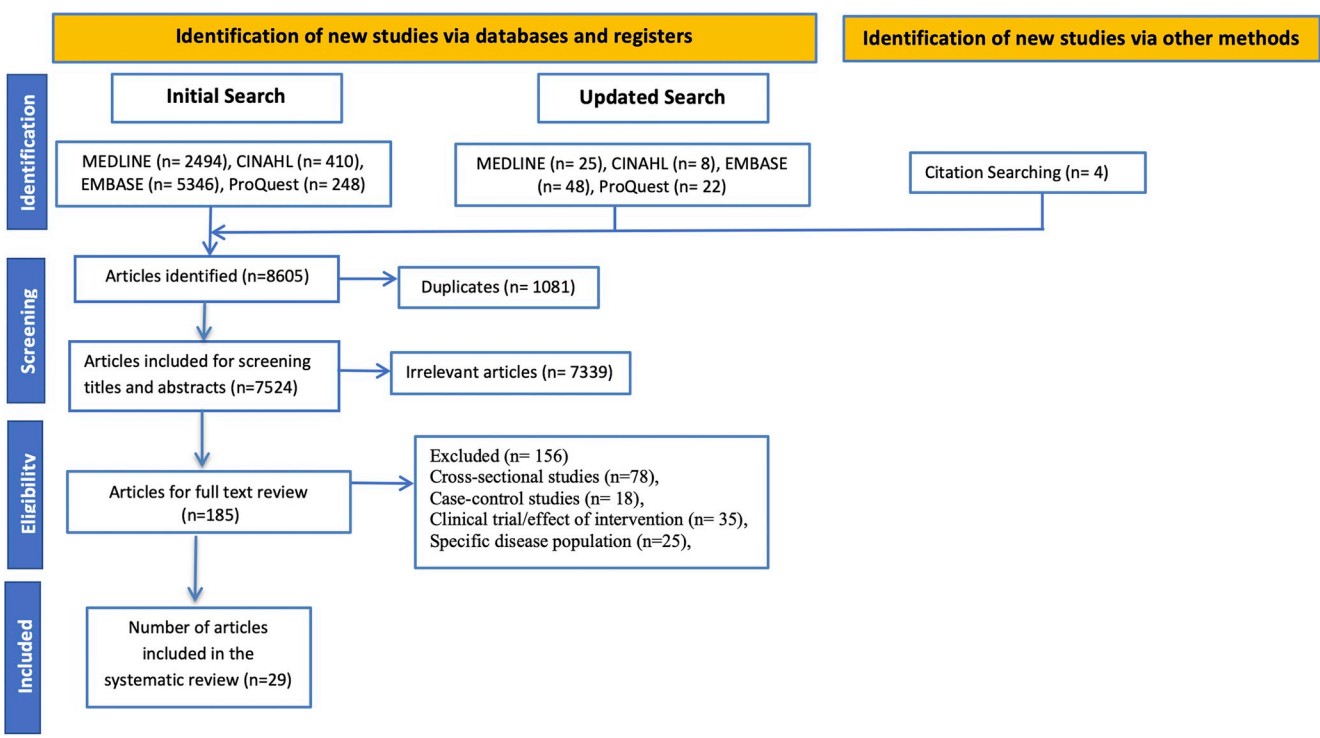

**Fig 1. Flowchart of study selection and inclusion.**

sorted into meaningful order based on the manner reported in the studies (e.g., low vs high). We did not restrict studies based on the length of follow-up but extracted that information for subsequent interpretation as a potential effect modifier.

Outcomes were limited to those broadly categorized as either pain (e.g., presence/absence of LBP or pain intensity) or disability (e.g., return to work or score on a standardized patient-reported outcome). Where studies reported "recovery" as an outcome those operationalizations were reviewed for relevance and if aligned with our purpose the verbatim definition was extracted and assigned to the most relevant outcome category (e.g., pain, disability, or both). The study protocol was prospectively registered in PROSPERO (registration no. CRD 42023402135) as part of a series of reviews on SDOHs and LBP.

## Data analysis and synthesis

Where possible, data were pooled and presented as odd ratios through random-effects meta-analysis using Comprehensive Meta-analysis software, version 2.2.04 (Biostat, Inc.©, Engle-wood, New Jersey). Syntheses were conducted for each of: i) onset of LBP (inception cohorts that start with no LBP and are followed over time to identify those who later report LBP); ii) pain intensity outcomes in acute LBP (inception starting within 8 weeks of LBP onset and followed over time to evaluate recovery), iii) pain-related disability outcomes in acute LBP, and iv) pain or disability outcomes in those entering the study with subacute (8 to 12 weeks) LBP. Heterogeneity in effects was assessed using both the I-squared statistic and p-value. I-squared <30% was deemed low heterogeneity, 30–60% as moderate, 61–75% as substantial, and 76–100% as considerable heterogeneity, and p-value at an alpha of <0.05 [19]. First, one estimate for pain and/or disability was calculated from unadjusted estimates as reported in each study. Where unadjusted (bivariate) estimates were not available, the adjusted estimates were pooled

and where enough data from primary sources were available, sensitivity analyses were conducted to determine whether adjustment for other covariates affected effect size estimates.

When heterogeneity could not be explained, or where there were too few primary sources to permit moderator analysis in otherwise highly heterogeneous effects, a narrative summary of the results is presented.

### GRADE assessment

Results across studies were synthesized using a modified Grading of Recommendations Assessment, Development and Evaluation (GRADE) approach that considered the strength of the effect (none, small, medium, or high) and confidence in the results (inconclusive, low, moderate, or high) based on RoB, precision, homogeneity and consistency of effects. Where effects could be statistically pooled, those results were used to determine effect size, where they could not, we used a qualitative synthesis approach focused on overall consistency across papers. For this review, we did not attempt to find study registration through online registries to identify publication bias as observational studies are not consistently registered and many studies were published prior to protocol registration becoming standard practice.

## Results

Fig 1 shows the PRISMA flow diagram. The search identified 8498 articles (including 1058 duplicates), of which 163 full texts were screened resulting in the inclusion of 23 articles. An additional 4 from reference lists and 2 from the update search were identified for a total of 29 manuscripts describing 27 prospective observational cohorts. The reliability between raters was Kappa = 0.97. Characteristics of the included studies are summarized in Table 1. The included studies were grouped into: onset of new LBP (n = 3), outcome of acute LBP (n = 18) and outcome of subacute LBP (n = 8). The publication date of the included studies spanned 1991 to 2022 and were from 13 countries. Sample sizes ranged from 53 [20] to 12,500 [21] and follow-up periods from 3 months (n = 4 [22–25]) to 3 years (n = 2 [21,26]).

### Risk of bias

Details of the RoB are reported in Table 1 and RoB for the overall body of literature is presented in Fig 2. The majority (n = 14) of the included studies were rated as high RoB with 13 rated moderate and 2 low. For the individual domains, low RoB was common in the domains of study participation (77%) and statistical analysis/reporting (74%). High RoB was common for the domains of confounding (45%) and study attrition (39%).

### Prognostic factors

Operational definitions and categories of EA were defined differently across studies. Thirteen studies [21,22,24,26,28,29,31,32,34,37,38,40,43] had EA according to a 12-year education grade (e.g., 12 years or less, 12 years or more). Three did not present adequate data for extracting years of education. Silva et al [36] categorized education as 'low, medium, or high" with no specific details, and Valencia et al [42] combined income and education into a single index of socio-economic status. Turner et al [41] reported EA as highest grade of education completed without detailing the years.

### Outcomes

Outcomes were broadly categorized into pain, disability or a combination of pain and disability. Pain was evaluated using 21 outcomes across 14 studies with the Numeric Pain Rating

**Table 1. Summary of findings: Educational attainment predicting the onset and outcomes in acute low back pain.**

| S/N | Author (year) Country | Aim | Sample and population | Categories of exposure (education) | Categories of outcome (pain or disability) | Design/ follow-up duration | Findings related to education | Study participation | Study attrition | Prognostic factor measurement | Outcome measurement | Confounding | Statistical analysis/ reporting | Overall Risk of Bias |
|---|---|---|---|---|---|---|---|---|---|---|---|---|---|---|
| **Onset of Back pain** | | | | | | | | | | | | | | |
| 1 | Zadro [27] 2017 Australia | Investigate whether EA affects the prevalence and risk of LBP differently in men and women while controlling for the influence of genetics and early shared environment. | 1,077 Adult monozygotic (MZ) and dizygotic (DZ) twins from the Murcia Twin Registry | Primary General secondary Superior secondary University | Activity limiting LBP questions: "Was this pain bad enough to limit your usual activities or change your daily routine for more than one day?" | 2 to 4 years | EA did not significantly affect the risk of LBP in the combined sample of men and women. Women with general secondary education (OR = 0.5, 95% CI: 0.2–0.9, p = .025), and University (adj OR = 0.3, 95% CI: 0.1–1.1, p = 0.66) have a lower risk of LBP compared to women with primary education. EA did not affect the risk of LBP in men. | L | L | M | M | M | L | M |
| 2 | Von Korff [26] 1993 USA | Assess whether depression was associated with increased risks of onset of 5 anatomically defined pain symptoms in a population-based, prospective study | 803 enrollees of a large health maintenance organization, Group Health Cooperative of Puget Sound (GHC). | High school or less Some college College graduate | Back pain: First onset of back pain and in prior 6 months of the baseline | 3-year | 3-year first onset probabilities of back pain, no significant differences between educational group (p>0.05) in both univariate and multivariate analysis. | L | L | L | L | H | L | H |
| 3 | George [28] 2012 USA | Identify variables from these domains that were predictive of occurrence of LBP, intensity, disability and psychological distress | 1230 consecutive subjects (18–35 years of age, without prior history of back pain) entering a training program at Fort Sam Houston, TX to become a combat medic in the U.S. Army were considered for participation from February 2007 to March 2008. | High school or lower, Some college, College or higher | In this study a prior history of LBP was operationally defined as a previous episode of LBP that limited work or physical activity, lasted longer than 48 hours, and caused the subject to seek health care. | 2 years. | High school or lower p = 0.204 High school or lower versus college or higher OR: 1.013 (0.65–1.58), p = 0.954 Highest pain intensity ratings for soldiers with high school or lower EA were 1.121 higher (95% CI =[0.239, 2.003]). | L | M | H | M | H | L | H |

*(Continued)*

**Table 1.** (Continued)

| S/N | Author (year) Country | Aim | Sample and population | Categories of exposure (education) | Categories of outcome (pain or disability) | Design/ follow-up duration | Findings related to education | Study participation | Study attrition | Prognostic factor measurement | Outcome measurement | Confounding | Statistical analysis/ reporting | Overall Risk of Bias |
|---|---|---|---|---|---|---|---|---|---|---|---|---|---|---|
| **Onset of Back pain** | | | | | | | | | | | | | | |
| **Acute Back Pain (<6weeks)** | | | | | | | | | | | | | | |
| 4 | Besen [25] 2015 USA | Develop and test a model of direct and indirect relationships among individual psychosocial predictors of return-to-work (RTW) outcomes following the onset of LBP. | 241 LBP patients with onset of less than 14 days followed from baseline to 3 months. | Education | Return to work | 3 months | Education was retained in the model, however for simplicity, we have omitted these paths from which shows the estimated model | M | M | L | L | L | M | M |
| 5 | Grotle [22] 2005 Norway | examine the clinical course of acute LBP and to evaluate prognostic factors for nonrecovery. | 123 patients with acute LBP 3 weeks consulting primary care for the first time were included, and 120 completed 3 months follow-up. | less or more than 12 years | Recovered if they scored ≤ 4 on the RMQ at both the 4 weeks and 3 months follow-up | 3 months. | Unadjusted OR = 0.59 (0.25–1.38), adjusted OR = 0.68 (0.28–1.69) Adjusting for age and sex | L | L | M | L | L | L | M |
| 6 | Grotle [29] 2007 Norway | Investigate the clinical course of pain and disability, and prognostic factors for non-recovery after 1-year, in patients seeking help for the first time due to acute LBP. | 123 (aged 18–60 years) first time acute LBP of <3 weeks duration, with or without radiating pain to the limb, no prior LBP treatment. | <12 years versus ≥12 years | Disability assessed by the RMDQ. Non-recovery was defined as an RMQ score of more than 4 | 1, 3, 6, 9, and 12 months. | High Education compared to low education for non-recovery at 12 months: crude OR = 0.65 (0.24–1.77); Adjusted (Age-and gender) OR: 0.71 (0.26–1.96), Adjusted for age and gender | L | H | L | L | H | L | H |
| 7 | Moffett [30] 2009 UK | Investigate whether socioeconomic status in patients with back pain participating in a randomized controlled trial was predictive of functional disability. | 949 secondary analysis was carried out only on UK BEAM (general practices in 14 centers around the UK with nonspecific LBP lasting at least 5 weeks) | Education up to (i) 16 years (or less), (ii) 17–19, (iii) over 20. | Roland Disability Questionnaire | 1, 3 and 12 months. | Disability (RDQ scores). Education level ≤16 vs. 17–19: 0.445 (0.21–0.942). Education level ≤16 vs. ≥20 0.364 (0.221–0.595). At 1, 3, and 12 months respectively. Adjusting for work status and Townsend score | L | M | L | M | L | L | M |

*(Continued)*

**Table 1.** (Continued)

| S/N | Author (year) Country | Aim | Sample and population | Categories of exposure (education) | Categories of outcome (pain or disability) | Design/ follow-up duration | Findings related to education | Study participation | Study attrition | Prognostic factor measurement | Outcome measurement | Confounding | Statistical analysis/ reporting | Overall Risk of Bias |
|---|---|---|---|---|---|---|---|---|---|---|---|---|---|---|
| **Onset of Back pain** | | | | | | | | | | | | | | |
| 8 | Starkweather [31] 2016 USA | Examine demographic, pain-related, psychological, and somatosensory characteristics in a cohort of acute LBP with persistent pain or unrecovered after 6 weeks. | 106 men and women between the age of 18 and 50 years diagnosed with an acute episode of LBP to participate | High school/ technical or lower; and Start college or higher | Recovered: if their pain resolved in the first 6 weeks after onset or as Persistent: if their pain continued for 6 months | 6 months | Persistent: High school = 13 (68%) v College = 6 (32%) Recovered: High school = 7 (24%), College = 22 (76) P = 0.0023 | L | M | L | L | L | L | M |
| 9 | Sterud [21] 2016 USA | Investigate the extent to which work-related factors contribute to the social gradient in LBP. | 12,550 randomly drawn cohort from the general population in Norway aged 18 to 66 years | Group 1: university/ college ≥4 years; Group 2: university/ college <4 years; Group 3: upper secondary; Group 4: incomplete upper secondary; Group 5: elementary secondary. | Reported intensity of LBP during the 4 weeks before answering the questionnaire: cases were defined as respondents who reported being severely or somewhat afflicted at follow-up | 3-year | All groups, OR: 0.609 95%CI: 0.524 to 0.708. | L | L | M | M | L | L | M |
| 10 | Menezes Coasta [32] 2009 Australia | Describe the course of chronic LBP in an inception cohort and to identify prognostic markers at the onset of chronicity. | 406 patients (aged 44.1±14.5 years) with recent onset chronic non-specific LBP. Participants were drawn from a larger cohort study of 973 consecutive patients with acute LBP (<2 weeks' duration) from primary care clinics in Sydney. | Categorize as no education beyond secondary school | Recovered if reported pain-free, and had no disability from LBP, and had returned to work for 30 consecutive days. | 12 months | No education beyond secondary school crude OR = 0.65 (0.48 to 0.88) p = 0.006, Adjusted OR = 0.74 (0.54 to 1.00), p = 0.05 Adjusted for previous sick leave due to LBP, risk of persistent pain, and pain related disability. Recovery from LBP related disability, No education beyond secondary school, crude OR = 0.72 (0.50 to 1.03), p = 0.07 | L | H | M | M | H | L | H |

*(Continued)*

**Table 1.** (Continued)

| S/N | Author (year) Country | Aim | Sample and population | Categories of exposure (education) | Categories of outcome (pain or disability) | Design/follow-up duration | Findings related to education | Study participation | Study attrition | Prognostic factor measurement | Outcome measurement | Confounding | Statistical analysis/reporting | Overall Risk of Bias |
|---|---|---|---|---|---|---|---|---|---|---|---|---|---|---|
| **Onset of Back pain** | | | | | | | | | | | | | | |
| 11 | Gurcay [23] 2009 Turkey | Assess the clinical course of patients with acute LBP throughout 12 weeks and to identify the prognostic factors for non-recovery in the short term. | 91 patients with acute LBP (<3 weeks), a mean of 37.9+10.3 years. | Education as years | Patients were considered recovered if they rated pain intensity as 0 and scored <4 on the RMDQ. | 1, 2, and 12 weeks. | Recovered group: 8.1±2.9 and non-recovered group: 8.1±3.6, p = 0.912 | L | H | L | L | H | M | H |
| 12 | Lehmann [33] 1993* USA | To develop a predictive model for occupational, physician and the spine consultant for high-risk long term disability. | 55 (37 men) patients aged between 18–65 (average 37.2 +9.07) years presenting with acute LBP who have incurred work absences of more than 2 but less than 6 weeks duration. | Education in years. None of the participants had a college degree. | Time until return to work | 6 months | No correlation between disabling LBP/ time until return to work and education (p = 0.17). | M | H | L | L | H | M | H |
| 13 | Gatchel [34] 1995 USA | Evaluate whether a comprehensive assessment of psychosocial measures is useful in characterizing those acute LBP patients who subsequently develop chronic pain disability problems. | 324 (ages: 35.2 ± 11.7 years) acute back patients of no more than 6 weeks. | No high school and High school versus Some college or more | LBP patients who were back at work at 6 months versus those who were not because of the original back injury | 6 months | These analyses revealed that the disabled group consisted of people tended to include those with lower EA, t (308) = 1.64, p < .10, relative to the nondisabled group. Higher education group: 123 Lower education group: 185 | L | H | M | L | H | L | H |
| 14 | Gatchel [35] 1995 USA | Evaluate the predictive power of psychosocial and personal factors in identifying acute LBP to progress to chronic pain disability | 421 patients (aged: 35.0 ± 11.5) seen at three clinics for an acute back pain episode of no more than 6 week | No high school and High school versus Some college or more | who were back at work at 12 months versus those who were not because of the original back injury | 12 months | No significant difference between disabled group and non-disabled group with education. | L | H | L | L | H | L | H |

(Continued)

**Table 1.** (Continued)

**Onset of Back pain**

| S/N | Author (year) Country | Aim | Sample and population | Categories of exposure (education) | Categories of outcome (pain or disability) | Design/follow-up duration | Findings related to education | Study participation | Study attrition | Prognostic factor measurement | Outcome measurement | Confounding | Statistical analysis/reporting | Overall Risk of Bias |
|---|---|---|---|---|---|---|---|---|---|---|---|---|---|---|
| 15 | Silva [36] 2022 Brazil | Describe pain and disability trajectories in older adults with a new episode of LBP. | 542 older adults, aged >55 years with a new episode of nonspecific LBP. Acute pain of <6 weeks and the individual has not sought care in health services owing to the LBP in the 6 months before the current. | Schooling level: low, medium and high | Pain intensity using NPRS and Disability (Roland-Morris Disability Questionnaire) | 12 months | Low schooling level (proportion and 95%CI): Pain trajectory: Pain recovery p = 0.03. Disability trajectories: | L | L | L | L | M | L | M |
| 16 | Seyedmehdi [37] 2015 Iran | Assess the power and quality of General Health Questionnaire (GHQ) for prediction of the odds of chronicity of acute LBP | 511 workers (500 were male) complaining of a new LBP in the past two weeks from large rubber factory in 2011–2013 in the city of Yazd, Iran | Educational level: <Diploma and ≥ Diploma | Chronic low back pain status: group 1 whose LBP lasted for <3 months and group 2 whose LBP lasted for ≥3 | 12 months | Unadjusted OR = 0.474 (0.294–0.758), p = 0.002 Adjusted OR = 0.73 (0.42735–1.235). Adjusted: age, sex, job parameters, smoking, BMI, exercise | L | L | L | M | L | L | M |
| 17 | Turner [38] 2008 USA | To identify early predictors of chronic work disability after work-related back injury. Interviewed three weeks after submitting a lost worktime. | 1885 (68% male) receiving work disability compensation from Washington State Department of Labor and Industries claims database July 2002 through April 2004, | -High school -Less than high school -Vocational or some college -College | The primary outcome was wage replacement compensation for temporary total disability ("work disability") 12 months after claim submission. | 12 months | Bivariate logistic regression analyses predicting 1-year work disability. Having a college degree was crude OR = 0.39 (0.21–0.75) but in the adjusted OR = 0.53 (0.23–1.18). | M | H | L | M | L L | L | H |
| 18 | Von Korff [39] 1991 USA | Improve understanding of the outcome of back pain in primary care patients. Three to six weeks after the index visit to the primary care clinic (via phone interview). | 1128 back pain patients with 3–6 weeks after an index visit of back pain recruited for back pain and follow up at 1 year. | College graduate HS graduate <12 years | Severity of back pain (0–10 ratings) and total days of back pain during a 6-month interval | 12 months | HS graduate versus college graduate: adj-OR = 1.85, p = 0.005 <12years versus college education: adj-OR = 3.17, p = 0.004 Adjusted: age, gender, education, race, pain grade, pain days, recency, disability payment | M | H | M | L | H | M | H |

(Continued)

**Table 1.** (Continued)

| S/N | Author (year) Country | Aim | Sample and population | Categories of exposure (education) | Categories of outcome (pain or disability) | Design/ follow-up duration | Findings related to education | Study participation | Study attrition | Prognostic factor measurement | Outcome measurement | Confounding | Statistical analysis/ reporting | Overall Risk of Bias |
|---|---|---|---|---|---|---|---|---|---|---|---|---|---|---|
| **Onset of Back pain** | | | | | | | | | | | | | | |
| 19 | Fliesser [40] 2018 Germany | Investigate associations between socioeconomic status indicators (education, job position, income, multidimensional index) and the genesis of chronic LBP. | 352 people with intermittent pain aged between 18 and 65 years of age, from four medical clinics across Germany as part of a national study on low back pain. Participants already reporting chronic pain syndrome were excluded. | International Standard Classification of Education from 0 (less than primary education) to 5 (tertiary education). | Chronic Pain Grade Questionnaires: Pain disability, pain intensity, pain class | 6-month | Pain disability Score Upper secondary education versus tertiary education B = 5.8, SE = 2.0, p <0.01** Pain intensity score. Upper secondary education versus tertiary education B: 4.3, SE: 2.0 p = 0.03* Adjusted for age and sex. | H | H | L | L | H | M | H |
| 20 | Turner [41] 2013 USA | Examine whether its predictive ability of the Chronic Pain Risk Score could predict risk for chronic pain | 521 adult patients initiating primary care for new episode of LBP from Group Health in Seattle. Patients who could not be reached within 14 days after this visit were excluded. | Highest grade of education completed. | Chronic pain risk model 4-month outcomes: chronic pain grade–algorithm incl. 3 pain severity and 3 pain interference. | 4 months | Expanded Chronic Pain Risk Model adjusted OR (95% CI): 0.48 (0.28, 0.81), p<0.01 Adjusting for all other variables: pain related outcomes, psychological opioid use | L | L | M | L | M | L | M |
| **Subacute (>6 weeks)** | | | | | | | | | | | | | | |
| 21 | Valencia [42] 2011 USA | Examined whether SES should be considered as mediator between the relationship of fear-avoidance beliefs and pain catastrophizing on disability, pain intensity and physical impairment | 108 patients with acute or sub-acute LBP, which were recruited from university of Florida affiliated orthopedic physical therapy clinics. | - 1) less than high school, 2) graduated from high school, 3) some college, 4) graduated from college, 5) some postgraduate, and 6) completed post graduate degree. | Pain intensity using NPRS and disability with ODI. | 1 and 6 months | SES did not predict pain intensity (Beta = -0.01, (std er: 0.01) P = 0.91) or disability (Beta = 0.75 (err: 0.62), P = 0.229) while controlling for pain catastrophizing SES correlated with pain catastrophizing (r = -0.37, P < 0.01). | H | H | M | L | H | M | H |

*(Continued)*

**Table 1.** (Continued)

| S/N | Author (year) Country | Aim | Sample and population | Categories of exposure (education) | Categories of outcome (pain or disability) | Design/ follow-up duration | Findings related to education | Study participation | Study attrition | Prognostic factor measurement | Outcome measurement | Confounding | Statistical analysis/ reporting | Overall Risk of Bias |
|---|---|---|---|---|---|---|---|---|---|---|---|---|---|---|
| **Onset of Back pain** | | | | | | | | | | | | | | |
| 22 | Turner [43] 2006 USA | Examine whether worker demographics, pain, disability, and psychosocial variables assessed soon after work-related LBP disability onset, predict 6-month work disability. | 1,068 workers, aged ≥18 years who submitted claims for work-related back pain and received at least 1 day of temporary total disability wage replacement (July 2002 to June 2003) | -Less than high school -High school -College or vocational/ technical -College graduate | Pain intensity in the past week NPRS Physical disability was assessed by the RDQ. | 6 months | Work disability Crude: - Less than high school 155 (15), OR: 0.99 (0.62–1.59) - High school 373 (35) OR: 1.00 - College or vocational/ technical 454 (42), OR: 0.86 (0.61–1.22) - College graduate 86 (8), OR: 0.46 (0.22–0.97). | L | M | L | L | L | L | M |
| 23 | Epping-Jordan [44] 1998 USA | Predictive relationship between pain intensity, disability and depressive symptoms | 78 men, 31.9±7 years of age and 12.8±1.6 years of education with first onset low back pain 8±2 weeks, who were part of a research program studying progression to chronic LBP. | Years of education | Pain intensity using the Descriptor Differential Scale, and disability with Sickness Impact Profile, | 2, 6, and 12 months | Correlation analysis: years of education were not significantly associated with pain intensity, disability, or depressive symptoms. | L | L | L | L | L | L | L |
| 24 | Melloh [20] 2011 New Zealand | Identify factors that influence the progression of acute LBP or recurrent LBP after a pain-free period of at least 6 months to the persistent state at an early stage. | 53 consecutive patients with acute LBP participated over the 6-month period | Higher and lower levels of EA | Patients suffering from persistent LBP by functional limitation that is disabling at baseline or 3-week follow-up (10 ODI points) and still severely impacts after 6 weeks | 6 months | OR = 0.491, 95%CI = 0.13 to 1.86 | M | L | M | L | L | M | M |
| 25 | Poiraudeau [24] 2006 France | Assess the outcome of subacute LBP to identify the characteristics of patients and physicians' beliefs about LBP. | 266 Rheumatologists and 440 patients with subacute low back pain (of 4–12 weeks only). | No full-time education Primary school High school Post-graduate | Persistence LBP: 'Has your low back pain persisted since your visit to your rheumatologist 3 months ago?' | 3 months | OR = 0.917, 95%CI = 0.58 to 1.29, P = 0.668 | L | M | L | M | H | L | H |

*(Continued)*

**Table 1.** (Continued)

| S/N | Author (year) Country | Aim | Sample and population | Categories of exposure (education) | Categories of outcome (pain or disability) | Design/follow-up duration | Findings related to education | Study participation | Study attrition | Prognostic factor measurement | Outcome measurement | Confounding | Statistical analysis/reporting | Overall Risk of Bias |
|---|---|---|---|---|---|---|---|---|---|---|---|---|---|---|
| **Onset of Back pain** | | | | | | | | | | | | | | |
| 26 | Williams [45] 1998 USA | Determine the extent to which job satisfaction predicts pain, psychological distress, and disability 6 months after an initial episode of LBP. | 136 consecutive men, aged between 18 and 50 years, which back pain (T6 or below) that had been present "on a daily previous 8 ±2 weeks. | Years of education | Pain (VAS) and disability (Sickness Impact Profile, Quality of Well-Being). | 2 and 6 months | Zero-order correlations among six-month outcome measures were non-significant with years of education, | L | M | L | L | L | L | M |
| 27 | Kongsted [46] 2015 Denmark | Identify LBP trajectories using LBP intensity and frequency measured once a week over 1 year and compare the results obtained using different analytical approaches. | 1,082 participants consulting a health care provider for the first time due to their current episode of LBP and must not have attended another chiropractor within the last 3 months | No formal post-high school education, vocational training, higher education of <3 years, higher education of 3–4 years, higher education of >4 years, | Eight subgroups to predict trajectory of back pain. From groups 1, Recovered, to group 8, moderate ongoing daily. | 2 weeks, 3 months, and 12 months | Subgroups were not significantly associated with EA. Subgroup 1 vs 5: OR = 0.5, p = 0.09 | L | M | L | M | M | M | M |
| 28 | Cats-Baril [47] 1991 USA | Develop a predictive risk model for low back pain disability better than experienced specialists and generalist. | 250 patients (aged,18–65 years) attending two secondary care for new episode of LBP, which has not prevented them from working for at least the last three months | Level of education | Whether a patient returns to work or becomes disabled, after 6 months | 3 and 6 months | The discriminate function analysis identified EA (p<0.05) as predictive of disabled | L | H | H | M | M | L | H |
| 29 | Karjalainen [48] 2003 Finland | Identify outcome determinants of subacute LBP. | 164 employed patients with subacute (duration of pain 4–12 weeks) daily LBP recruited from 36 primary health care centers in the Helsinki metropolitan area | High school diploma | Sick leave because of back pain, intensity of pain (rated 0 to 10), back-specific perceived disability (ODI). | 3, 6, and 12 months. | Univariate: Intensity of pain = p<0.01 ODI = p = 0.57 Multivariate: intensity of pain = p>0.05 Adjusted for all significant values in the first model: Age, gender, occupation, health status. | L | L | L | L | L | L | L |
| | Key: | | L: Low Risk of Bias | M: Moderate Risk of Bias | H: High Risk of Bias | | | | | | | | | |

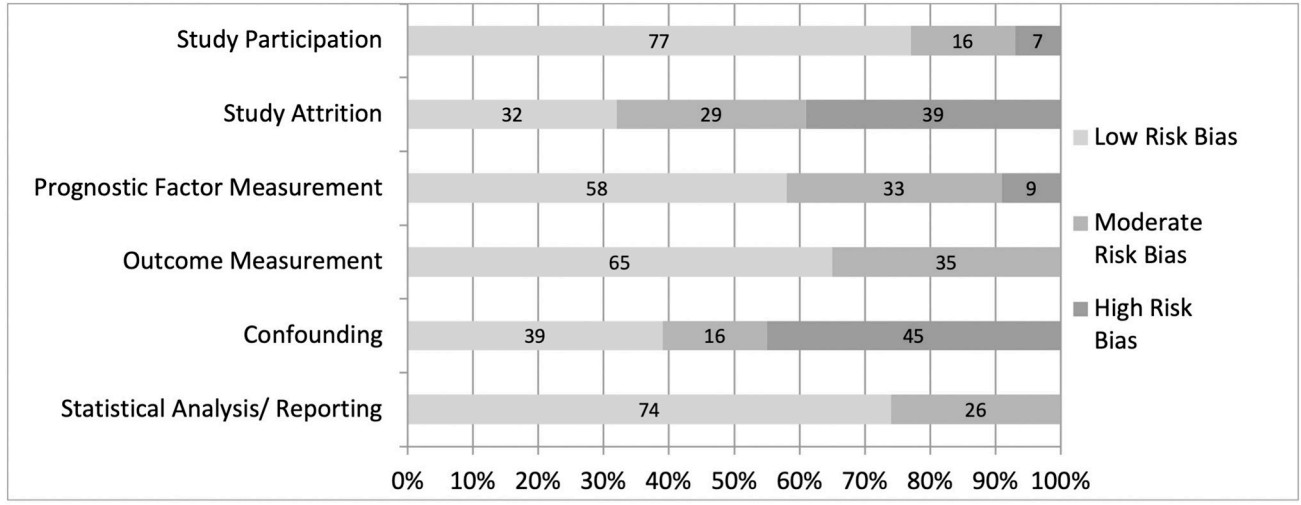

**Fig 2. Percentage distribution of the types of bias in the quality appraisal criteria of the included studies.**

Scale (NPRS), [29,36,40,42,43] and Visual Analog Scale (VAS) [45] the most frequent. Disability was evaluated using 21 outcomes in 9 studies most commonly using the Oswestry Disability Index (ODI) [20,42,48] and Roland Morris Disability Questionnaire (RMDQ) [23,29,36]. A combination of both pain and disability was evaluated in two studies [23,32].

## Meta-analysis

Of the 29 articles from 27 studies, 9 articles [25,33,35,39,44–48] did not report adequate data (e.g., proportions, estimates) to permit statistical pooling and we were unsuccessful in contacting the original authors of those papers. Accordingly, data from 19 studies were available for meta-analysis.

**EA as a predictor for the onset of new LBP.** Fig 3 shows the forest plot for pooled estimate for the association between EA and the onset of LBP. Three longitudinal inception cohort studies [26–28] (total N = 3,110) presented adequate data for pooling. The pooled effect of the three studies shows consistent evidence of no association between EA and new onset of LBP (OR = 0.93; 95%CI 0.75 to 1.15) with homogeneity ($I^2 < 0.1\%$).

**EA as a prognostic variable for predicting outcome of acute LBP.** Eight studies (total N = 15,079) reporting pain as an outcome were pooled with low-moderate heterogeneity in effect sizes ($I^2 = 35\%$). Results supported a significant effect, in which higher EA predicts lower LBP symptoms 3 months to 3 years after onset of acute LBP (OR = 0.54; 95%CI 0.44 to 0.67, Fig 3). Three of those studies reported adjusted estimates only, excluding those resulted in an equivocal shift in pooled effects using only the unadjusted estimates (OR = OR: 0.51, 95%CI: 0.37 to 0.70, I = 36%, Fig 4). Nine articles (8 cohorts, total N = 4,672 subjects) reported a pain-related disability outcome. Pooling similarly indicated that higher EA measured in the acute phase of LBP predicts lower pain-related disability 3 months to 12 months later (OR = 0.57; 95%CI 0.42 to 0.76, Fig 3) with moderate heterogeneity ($I^2 = 44\%$). Excluding the adjusted estimates from two of those studies again resulted in an equivocal shift in pooled effect (OR: 0.62, 95%CI: 0.51 to 0.77, Fig 4) but without heterogeneity ($I^2 = 0\%$). Of the three studies that could not be pooled, two studies [25,39] (moderate RoB, total N = 1,369) also reported significant associations between outcomes of acute LBP and EA. The third [33] (1 moderate RoB, N = 55 subjects) reported no association between EA and disabling LBP/time to return to work.

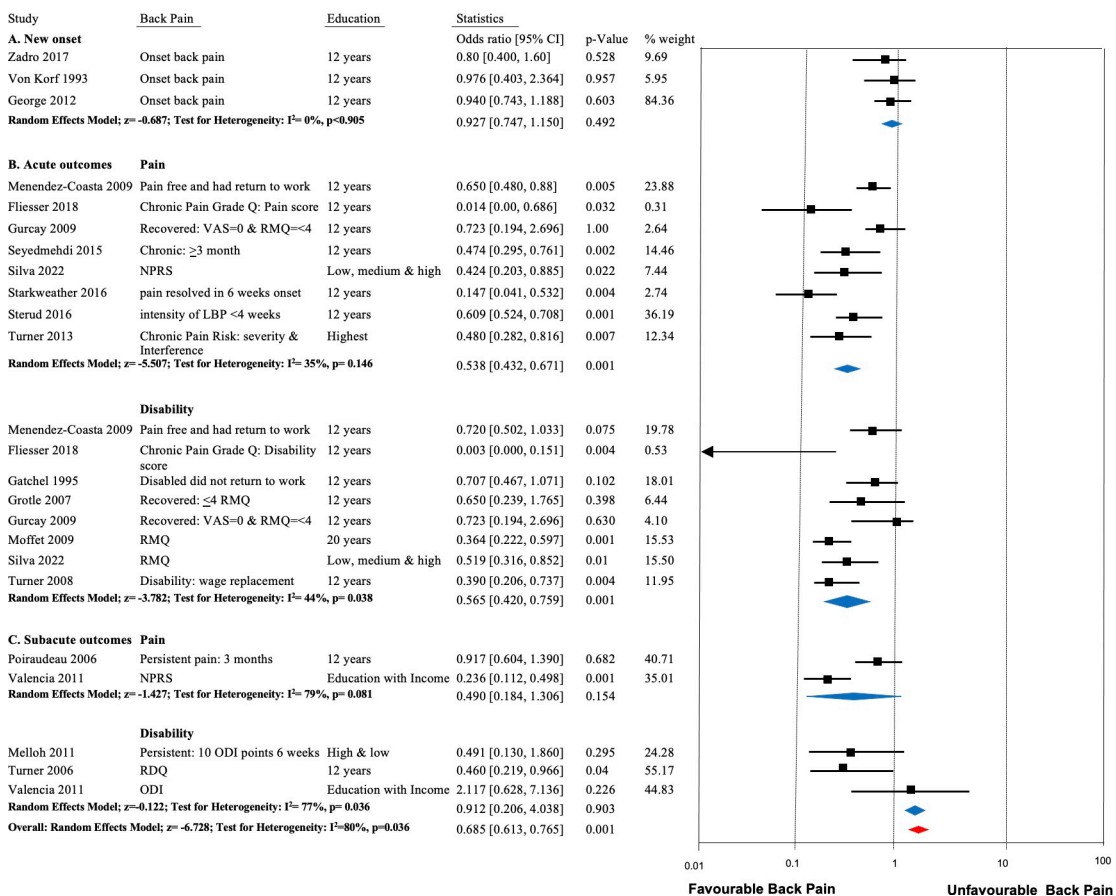

**Fig 3.** Forest plot of prognostic accuracy (odds ratio, OR) of educational attainment for predicting a) new onset, b) acute outcomes, and c) subacute outcomes in LBP.

**EA as a prognostic variable for predicting outcome of subacute LBP.** Two (pain severity) [24,42] and three (pain-related disability) [20,42,43] studies predicting outcomes in subacute LBP could not be meaningfully pooled owing to high heterogeneity ($I^2 > 79\%$), inconsistent outcomes, and too few sources to permit moderator analysis. Accordingly, we proceeded with qualitative synthesis. For pain intensity as an outcome, 4 of 6 studies (2 low [44,45] and 2 moderate [24,46] RoB, total N = 2,880) reported unadjusted (bivariate) estimates and indicated no association between EA and follow-up outcome. The remaining two studies (1 low [48] and 1 moderate [42] RoB, total N = 272) reported no significant association after adjusting for pain catastrophizing [42] or age, gender, occupation, and health status [48]. For pain-related disability, 7 studies (1 low, 3 moderate and 2 high RoB studies, total N = 1,960) reported inconsistent evidence. Two studies [43,47] (total N = 1,504) found a significant negative association between EA and pain-related disability as measured with the RDQ [43] and ability to return to work [47]. Four other studies [42,44] (2 low, [20,48] 1 moderate [20] and 1 high [42] RoB, total N = 403) reported no association between EA and ODI [20,42,48] or sickness profile [44].

**Sex based analysis of educational attainment and low back pain outcomes.** Two studies analyzed data for potential differential effects of EA on LBP when disaggregated by sex. The two studies could not be pooled due to differences in case definitions. Zadro et al [27] studied new onset LBP and reported lower EA to be associated with increased proportion of new onset

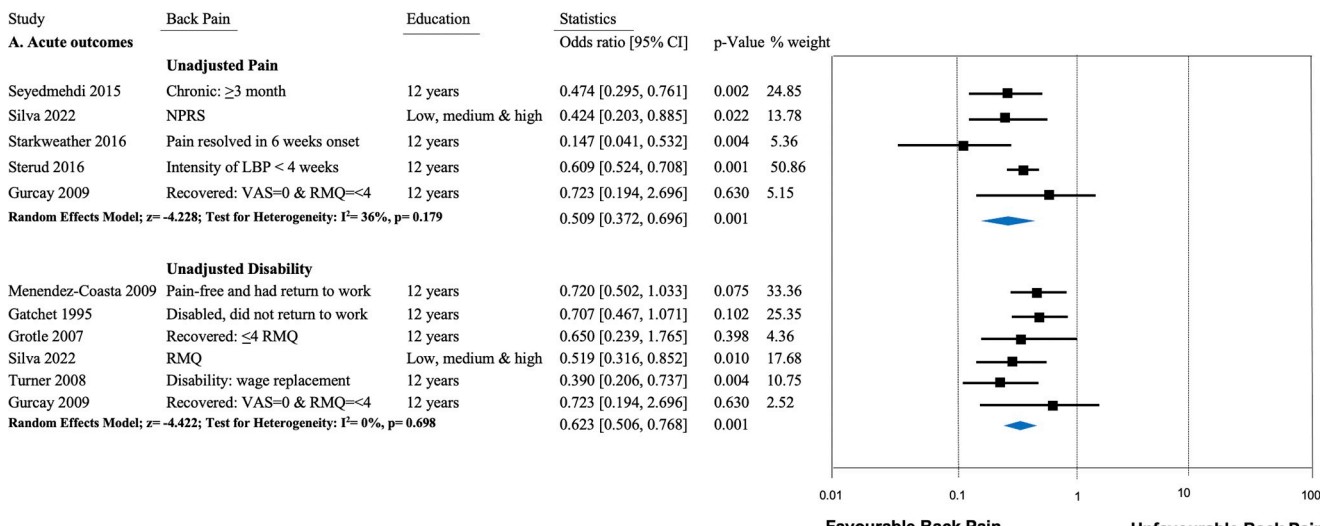

**Fig 4. Forest plot of the association of unadjusted back pain outcomes (acute and subacute) and educational attainment with pooled estimates and 95% confidence intervals.**

LBP in females only, with no significant effects in males. Sterud et al [21] evaluated outcomes in acute LBP and found no differential effect on outcomes between sexes.

## GRADE statement

Evidence profile of all included studies applied using GRADE is presented in Table 2.

**EA and onset of LBP**: On the basis of 3 studies, 1 moderate and 2 high RoB, with consistency in magnitude of effect, we find low-to-moderate confidence that EA has no association with the onset of new LBP in adults when followed for at least 2 years.

**EA and outcomes of acute LBP**: On the basis of 9 of 11 studies, 6 moderate and 3 high RoB, we have moderate confidence that EA when collected at inception shows a significant association with LBP symptom severity measured at least 3 months later. We find low confidence based on 5 of 11 studies, 2 moderate and 3 high RoB, of a similar association when the outcome is pain-related disability.

**EA and outcomes of subacute LBP**: On the basis of 7 of 9 studies, 2 low, 3 moderate, and 2 high RoB, we find inconsistent evidence and very low confidence in any association between EA and subsequent outcomes of pain severity when participants are incepted at the subacute (8–12 weeks from onset) stage. We find inconsistent evidence based on 3 of 5 studies (1 low, moderate and high RoB) and very low confidence for no association between EA and disability at least 3 months later. Significant heterogeneity in case definitions and effect sizes preclude more definitive findings.

## Discussion

We have conducted a rigorous and systematic search, extraction, and pooling of effects to explore the associations between a key indicator of socioeconomic status (highest level of education completed) and each of onset of new LBP, outcomes of acute LBP, or outcomes of subacute LBP. This represents part of an ongoing set of reviews exploring the social determinants of health and their associations with LBP, conceptualized herein as evaluating whether lower EA (high school or less) functions as either a potential risk or prognostic factor for onset or

**Table 2. Evidence profile for the association between low back pain outcomes and educational attainment.**

| Low back pain and educational attainment findings | GRADE Variables | | | | | | |
|---|---|---|---|---|---|---|---|
| STUDIES | Association | Risk of Bias | Inconsistency | Indirectness | Imprecision | Publication bias | Certainty in the evidence* |
| *Onset* | | | | | | | |
| Zadro [27] Von Korff [26] George [28] | x | M = 1 H = 2 | No Serious inconsistency | No Serious indirectness | No serious imprecision | Unlikely | ⊕⊕⊕⊕ High certainty for evidence of no association between educational attainment and onset |
| *Acute pain* | | | | | | | |
| *Besen [25] *Karjalainen [48], Starkweather, [31] Sterud [21], Silva, [36] Seyedmehdi [37], *Von Korff [39], Fliesser [40] Turner, [41] Coasta [32] | - | H = 3 M = 6 | Moderate inconsistencies | No Serious indirectness | No serious imprecision | Unlikely | ⊕⊕⊕O Moderate certainty of association between low educational attainment and poor pain outcome in acute LBP |
| Coasta [32], Seyedmehdi [37], | x | H = 1 M = 1 | | | | | |
| *Acute disability* | | | | | | | |
| Moffett [30], Silva [36] Turner [38] Fliesser [40] Coasta [32] | - | M = 2 H = 3 | Moderate inconsistencies | No Serious indirectness | No serious imprecision | Unlikely | ⊕⊕OO Low certainty of evidence for the association between low educational attainment and poor disability outcomes in acute LBP |
| Coasta [32], Gatchel [34,35] Grotle [22,29], Gurcay [23] Turner [38], *Lehmann [33] | x | H = 5 L = 1 | | | | | |
| *Subacute pain* | | | | | | | |
| Valencia [42], Karjalainen [48] | - | L = 1 H = 1 | No Serious inconsistency | Moderate indirectness | No serious imprecision | Unlikely | ⊕⊕⊕O Moderate certainty of evidence of no association between educational attainment and pain outcome in subacute LBP |
| Valencia [42], *Epping-Jordan, [44] Poiraudeau [24] *Williams [45] *Kongsted [46], Karjalainen [48] | x | L = 2 M = 2 H = 2 | | | | | |

*(Continued)*

**Table 2.** (Continued)

| Low back pain and educational attainment findings | GRADE Variables | | | | | | |
|---|---|---|---|---|---|---|---|
| STUDIES | Association | Risk of Bias | Inconsistency | Indirectness | Imprecision | Publication bias | Certainty in the evidence* |
| *Subacute disability* | | | | | | | |
| Turner [43] *Cats-Baril [47] | - | M = 1 H = 1 | Serious inconsistency | Moderate indirectness | No serious imprecision | Unlikely | ⊕⊕OO Very low certainty of evidence for the association between educational attainment and disability outcome in subacute LBP |
| Valencia [42], *Epping-Jordan [44] Karjalainen [48], Melloh, [20] | x | H = 1 M = 1 L = 2 | | | | | |

outcome of LBP, respectively. As observational studies, these are inherently vulnerable to confounding bias from several other potential variables meaning causation should not be assumed. Based on the strength and effects of available evidence, we have moderate confidence in a significant negative *association* between EA and pain severity or disability outcomes of acute LBP in which higher EA may offer some protection against poor outcomes, low confidence that EA has no association with onset of new LBP, and very low confidence of any association between EA and outcome when starting from the subacute LBP stage.

While to our knowledge, the quantitative synthesis of evidence related to new-onset LBP is novel, our results are largely consistent with those of other reviews in acute or subacute LBP, each of which included EA as part of a larger set of potential prognostic variables and few of which conducted meta-analyses. Previous LBP studies have failed to establish an association between EA and LBP outcomes for various reasons such as a small sample size to statistically power the study to detect effects [43], lack of uniform study design [49] and heterogeneous population, among others. For example, a review by Batista et al [49] reported that people with higher EA are less often affected by the occurrence of LBP. However, that review included multiple study designs that might have added statistical noise to the estimates of effect. Cancelliere and colleagues [50] based on best evidence synthesis of systematic reviews on factors affecting return to work after injury or illness identified higher EA and socioeconomic status among factors associated with positive return-to-work outcomes. A similar review by Dionne and colleagues [51] included multiple study designs that could not allow established prediction or causation, though that review concluded that people with lower EA are more likely to be affected by disabling LBP.

While it is tempting to ascribe mechanisms to our results, any such attempt is necessarily speculative given the design of studies and the inability to feasibly conduct a randomized trial in which one arm remains uneducated. Accordingly, criteria to support cause-and-effect, most famously described by Bradford-Hill [52,53] may never be fully realized. However, it also limits the impact of this work if no potential mechanisms are explored. EA is commonly included as part of the indices used to assign people to socioeconomic strata [54,55], that also include variables such as annual household income and median neighborhood income. From a

Bourdieusian perspective, each of these may be interpreted as inferring capital that can be converted to power across different social fields [56]. In the context of outcomes of LBP, possessing social capital enabled by higher EA may permit easier access to effective care or alternative employment options, meaning that research using outcomes such as work status may find those with economic or educational privilege have better outcomes. However, EA may also be functioning more as a proxy for other influences on experiences of health and wellness outcomes. For example, higher EA may signal higher health literacy, living in more affluent areas with easier access to schools, or family wealth. Lower EA may be associated with, amongst other things, experiences of school bullying, early parentification, poor mental health, or neighborhood poverty [57]. Each may also play a moderating or mediating role on health outcomes [58], suggesting that these effects are very likely complex interactions between person- and society-level influences.

That EA showed no significant association with the onset of new LBP also demands further interpretation. Importantly, on the basis of only 3 studies of moderate-to-high RoB, we cannot have more than low confidence in the finding, though the consistency from over 3,000 participants is meaningful. Intuitively we might expect that those with lower EA are also more likely to be in jobs that demand higher physical labour or more repetitive tasks that might increase the risk of musculoskeletal disorders like LBP. However, we can see prior evidence that appears to support the lack of association identified herein. For example, in a large population-level study of >74,000 U.S. adults aged 30–49, Zajacova and colleagues found a non-linear association between EA and pain, in which adults who started but did not finish a post-secondary educational program reported a higher prevalence of painful conditions than either those with completed post-secondary education *or* those with secondary education only [59]. There is also an abundance of evidence associating sedentariness or prolonged sitting, as may be more likely experienced by those with higher-level or managerial roles, as risk factors for low back pain. Further, amongst blue-collar workers Lagersted-Olsen and colleagues found no association between daily time spent in a forward-bending posture and onset or aggravation of LBP over one year [60]. Accordingly, similar to our commentary on EA and LBP outcomes, any association between EA and LBP onset is likely complex and it seems overly simplistic to suggest that lower education does or does not lead to LBP.

Limitations to the study include the inability to establish causation as previously described, though this is more a limitation of the overall field rather than this particular review. With very few exceptions we were also largely unable to retrieve missing or under-reported data by contacting authors, meaning that some potentially relevant data have not been included in the meta-analyses that may otherwise change the results. We did not include studies published in a language other than English or without a formal English translation available, raising the risk that we have missed data from work published in non-English journals that may influence our results. Additionally, due the limited number of studies (fewer than 10) included in the meta-analysis, analysis of publication bias may be inappropriate [61]. Finally, not all studies reported EA in a way that permitted easy dichotomization into the 12-year categories. We made our best estimates based on reporting in the manuscript when grouping results into one of these two categories, though acknowledge that some errors may have been made.

While this review suggests EA is not associated with onset of new LBP or outcomes of subacute LBP, it does suggest a consistent association with outcomes of acute LBP. We have proposed potential mechanisms to explain these findings, though clearly more theoretical and empirical work is needed in this field. Future high-quality longitudinal studies with adequate sample size, clear and consistent definitions of EA and adjusting for meaning confounders in the study design and/or analysis will improve the understanding of the relative contribution of EA to the onset, and outcome, of acute and subacute LBP.

## Supporting information

**S1 Checklist. PRISMA 2020 checklist.**
(DOCX)

**S1 Appendix.**
(DOCX)

## Author Contributions

**Conceptualization:** David M. Walton.

**Data curation:** Aliyu Lawan, Alex Aubertin, Jane Mical, Joanne Hum, Michelle L. Graf, Peter Marley, Zachary Bolton, David M. Walton.

**Formal analysis:** Aliyu Lawan, Alex Aubertin, Jane Mical, Joanne Hum, Michelle L. Graf, Peter Marley, Zachary Bolton, David M. Walton.

**Investigation:** Aliyu Lawan, David M. Walton.

**Methodology:** Joanne Hum, Zachary Bolton, David M. Walton.

**Project administration:** David M. Walton.

**Resources:** David M. Walton.

**Supervision:** David M. Walton.

**Validation:** David M. Walton.

**Visualization:** Alex Aubertin, Jane Mical, Joanne Hum, Michelle L. Graf, Peter Marley, Zachary Bolton.

**Writing – original draft:** Aliyu Lawan, Alex Aubertin, Jane Mical, Joanne Hum, Michelle L. Graf, Peter Marley, Zachary Bolton.

**Writing – review & editing:** Aliyu Lawan, Alex Aubertin, Jane Mical, Joanne Hum, Michelle L. Graf, Peter Marley, Zachary Bolton, David M. Walton.

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
