## [Decision Letter · Decision Letter 0]

2 Jun 2024

PONE-D-24-08660Is educational attainment associated with the onset and outcomes of low back pain? A systematic review and meta-analysisPLOS ONE

Dear Dr. Lawan,

Thank you for submitting your manuscript to PLOS ONE. After careful consideration, we feel that it has merit but does not fully meet PLOS ONE’s publication criteria as it currently stands. Therefore, we invite you to submit a revised version of the manuscript that addresses the points raised during the review process.

We look forward to receiving your revised manuscript.

Kind regards,

Mohammad Ali

Academic Editor

PLOS ONE

2. We note that your Data Availability Statement is currently as follows: [The data underlying the results presented in the study are available within the manuscript.]

Reviewers' comments:

Reviewer's Responses to Questions

**Comments to the Author**

1. Is the manuscript technically sound, and do the data support the conclusions?

Reviewer #1: Yes

Reviewer #2: Yes

2. Has the statistical analysis been performed appropriately and rigorously? 

Reviewer #1: N/A

Reviewer #2: Yes

3. Have the authors made all data underlying the findings in their manuscript fully available?

Reviewer #1: Yes

Reviewer #2: Yes

4. Is the manuscript presented in an intelligible fashion and written in standard English?

Reviewer #1: Yes

Reviewer #2: Yes

5. Review Comments to the Author

Reviewer #1: This is a systematic review with meta-analysis on a very relevant topic, both due to the high worldwide prevalence of the outcome (LBP) and the significant importance of the exposure (educational attainment). Therefore, acquiring knowledge about the association between educational attainment and LBP is extremely necessary. However, despite generally following the recommendations of PRISMA 2020 (whose updated reference was not provided), some items related to the quality of SR advocated by the AMSTAR 2 tool were not met. For example, I missed the inclusion of articles in languages other than English, or at least a plausible justification for the exclusion of other languages. It was also not stated whether experts in the field were sought for suggestions on the inclusion of other studies. The PRISMA flowchart presented is not the currently recommended. The recommendation of an important item from AMSTAR 2, item 7 (Did the review authors provide a list of excluded studies and justify the exclusions?), was not fulfilled. The same applies to items 15 (If they performed quantitative synthesis did the review authors carry out an adequate investigation of publication bias (small study bias) and discuss its likely impact on the results of the review?) and 16 (Did the review authors report any potential sources of conflict of interest, including any funding they received for conducting the review?) of AMSTAR 2. Finally, regarding the discussion, I felt that the explanation of the strong points of this systematic review with meta-analysis and future proposals for development was missing.

Reviewer #2: I read the manuscript entitled "Is Educational Attainment Associated with the Onset and Outcomes of Low Back Pain? A Systematic Review and Meta-analysis" with great pleasure. The article holds excellent value for scientific and clinical audiences, adding to the evidence on the association between socioeconomic status and low back pain. It is well-written, concise, and technically sound.

I've made a few minor comments which could help to improve the manuscript further.

First, when reading the article, I needed some clarification on the specific eligibility criteria used to include articles in the systematic review. Are those the ones mentioned in the paragraph "Design", i.e., observational prospective cohort or population-level studies (not clinical trials) of patients aged >18 years with either no LBP at inception (followed to determine onset), or acute (<8 weeks) and/or sub-acute (8-12 weeks) LBP. Did you, for instance, also have specific criteria for the methods for evaluating educational attainment, the outcomes of low back pain, the language in which the articles were written, or the fact that the articles had to have been peer-reviewed?

Second, I needed help understanding the sentence with lines 199 - 201. Could you further improve this sentence?

Finally, there were some minor typos or missing words throughout the manuscript, which could be corrected.

6. PLOS authors have the option to publish the peer review history of their article (what does this mean?). If published, this will include your full peer review and any attached files.

Reviewer #1: **Yes: **CARLOS AUGUSTO FERREIRA DE ANDRADE

Reviewer #2: No

---

## [Author Response · Author response to Decision Letter 0]

13 Jun 2024

The Academic Editor, 

PLOS ONE

Response to Editors & Reviewer Comments: Is educational attainment associated with the onset and outcomes of low back pain? A systematic review and meta-analysis (PONE-D-24-08660).

The authors sincerely thank the Academic Editor for finding our work worthy of consideration in PLOS ONE and for the opportunity to revise our submission. Also, we appreciate the reviewers for their thoughtful comments and suggestions to improve the manuscript. We have addressed the comments raised as follows: 

Comment: 1. Please ensure that your manuscript meets PLOS ONE's style requirements, including those for file naming. The PLOS ONE style templates can be found at

Response: The manuscript and all accompanying files are in accordance with PLOS ONE’s style requirements, and all named accordingly. 

Comment: 2. We note that your Data Availability Statement is currently as follows: [The data underlying the results presented in the study are available within the manuscript.]

Response: We are confirming that the submission contains all raw data required to replicate the results of your study, as stated on the title page line 20 to 21 “Data Availability Statement: The raw data involved in this analysis and required to replicate the results are available within the manuscript.”

Comment: 3. Please include captions for your Supporting Information files at the end of your manuscript, and update any in-text citations to match accordingly. Please see our Supporting Information guidelines for more information: http://journals.plos.org/plosone/s/supporting-information.

Response: There are no supporting documents needed to be added to the submission. 

Reviewers' comments:

Reviewer's Responses to Questions

Comments to the Author

1. Is the manuscript technically sound, and do the data support the conclusions?

Reviewer #1: Yes

Reviewer #2: Yes

Response: Thank you for agreeing with the technical quality of our manuscript, including that the data support the conclusion made. 

2. Has the statistical analysis been performed appropriately and rigorously?

Reviewer #1: N/A

Reviewer #2: Yes

Response: As agreed, the data undertake a meta-analysis using random effect to explore the predictive value of educational attainment and low back pain. Sensitivity analysis was undertaken to explore the effect of moderating variables as recommended. While variables with high heterogeneity were analyzed qualitatively. A summary of evidence was presented using GRADE.

3. Have the authors made all data underlying the findings in their manuscript fully available?

Reviewer #1: Yes

Reviewer #2: Yes

Response: We have confirmed as well that all data needed to replicate the findings are here within the manuscript.

4. Is the manuscript presented in an intelligible fashion and written in standard English?

Reviewer #1: Yes

Reviewer #2: Yes

Response: Despite the unanimous agreement between the reviewers on the intelligible fashion and standard language used to write the manuscript. We have thoroughly revised the manuscript for any typographical or grammatical errors and have made minor revisions to improve readability throughout.

5. Review Comments to the Author

Reviewer #1: This is a systematic review with meta-analysis on a very relevant topic, both due to the high worldwide prevalence of the outcome (LBP) and the significant importance of the exposure (educational attainment). Therefore, acquiring knowledge about the association between educational attainment and LBP is extremely necessary. However, despite generally following the recommendations of PRISMA 2020 (whose updated reference was not provided), some items related to the quality of SR advocated by the AMSTAR 2 tool were not met. For example, I missed the inclusion of articles in languages other than English, or at least a plausible justification for the exclusion of other languages. It was also not stated whether experts in the field were sought for suggestions on the inclusion of other studies. The PRISMA flowchart presented is not the currently recommended. The recommendation of an important item from AMSTAR 2, item 7 (Did the review authors provide a list of excluded studies and justify the exclusions?), was not fulfilled. The same applies to items 15 (If they performed quantitative synthesis did the review authors carry out an adequate investigation of publication bias (small study bias) and discuss its likely impact on the results of the review?) and 16 (Did the review authors report any potential sources of conflict of interest, including any funding they received for conducting the review?) of AMSTAR 2. Finally, regarding the discussion, I felt that the explanation of the strong points of this systematic review with meta-analysis and future proposals for development was missing.

Response: 

1. The updated reference for PRISMA 2020 is now included in the citation. 

2. We have included all language of publication as stated in the selection criteria (line: 148). While our search did not return any hits that appeared relevant but were written in another language, we also acknowledge that the databases we searched tend to favour English-language journals. This work is part of a larger body of research led by leading academics in the field of spinal (low back and neck) pain who are well-versed in the field and are unaware of other citations missed here, though admittedly that offers only a modicum of confidence in the exhaustiveness of our yield. The search strategy was extensively developed and applied with the help of a Health Science librarian to maximize the likelihood that all relevant studies are captured. While is seems unlikely that important work has been missed, we have nonetheless alluded to this potential limitation with the following addition to the Limitations section: “We did not include studies published in a language other than English or without a formal English translation available, raising the risk that we have missed data from work published in non-English journals that may influence our results.”.

3. The PRISMA flowchart is now updated according to the recommended PRISMA 2020 guideline (Page et al 2021).

4. The recommendation of an important item from AMSTAR 2, 

a. Item 7 (Did the review authors provide a list of excluded studies and justify the exclusions?), was not fulfilled. 

Response: The updated PRISMA flowchart included the list of reasons and the number of studies excluded (see Fig 1Flow diagram.tif).

b. Item 15 (If they performed quantitative synthesis did the review authors carry out an adequate investigation of publication bias (small study bias) and discuss its likely impact on the results of the review?) and 

Response: Acknowledging that assessing publication is a difficult task in meta-analysis with few studies. According to Cochrane’s Recommendations on testing for funnel plot asymmetry: “As a rule of thumb, tests for funnel plot asymmetry should be used only when there are at least 10 studies included in the meta-analysis, because when there are fewer studies the power of the tests is too low to distinguish chance from real asymmetry” (https://handbook-5-1.cochrane.org/chapter_10/10_4_3_1_recommendations_on_testing_for_funnel_plot_asymmetry.htm).

This considering that we did not have any meta-analysis that included at least 10 studies, we are unable to perform a funnel plot to test for publication bias and this is noted in as one of the study’s limitations on page 24 line 449 to 450. “Additionally, due the limited number of studies (fewer than 10) included in the meta-analysis, analysis of publication bias may be inappropriate.(Lau et al., 2006)“

 In addition, we have included on the meta-analysis forest plot, percent weight for each study that is based on the effect sizes and sample sizes. The percent weights are automatically calculated and adjusted in the relative contribution of each study to the overall pooled estimate. The weight assigned to each study typically reflects the precision of the study's effect estimate, which is influenced by several factors including sample size. Larger studies usually provide more precise estimates and thus are given more weight compared to smaller studies. This may further help the interpretability of the results in the absence of funnel plot for the publication bias. 

c. Item 16 (Did the review authors report any potential sources of conflict of interest, including any funding they received for conducting the review?) of AMSTAR 2. 

Response: According to the PLOS ONE requirement, the authors declare no conflict of interest with the nature of the work described, including no funds received for this review as contained on the title page under Disclosure; line 19.

5. Finally, regarding the discussion, I felt that the explanation of the strong points of this systematic review with meta-analysis and future proposals for development was missing.

Response: Implication for future studies proposal has been added as a final paragraph to the manuscript: “While this review suggests EA is not associated with onset of new LBP or outcomes of subacute LBP, it does suggest a consistent association with outcomes of acute LBP. We have proposed potential mechanisms to explain these findings, though clearly more theoretical and empirical work is needed in this field. Future high-quality longitudinal studies with adequate sample size, clear and consistent definitions of EA and adjusting for meaning confounders in the study design and/or analysis will improve the understanding of the relative contribution of EA to the onset, and outcome, of acute and subacute LBP.”

Reviewer #2: I read the manuscript entitled "Is Educational Attainment Associated with the Onset and Outcomes of Low Back Pain? A Systematic Review and Meta-analysis" with great pleasure. The article holds excellent value for scientific and clinical audiences, adding to the evidence on the association between socioeconomic status and low back pain. It is well-written, concise, and technically sound.

I've made a few minor comments which could help to improve the manuscript further.

First, when reading the article, I needed some clarification on the specific eligibility criteria used to include articles in the systematic review. Are those the ones mentioned in the paragraph "Design", i.e., observational prospective cohort or population-level studies (not clinical trials) of patients aged >18 years with either no LBP at inception (followed to determine onset), or acute (<8 weeks) and/or sub-acute (8-12 weeks) LBP. Did you, for instance, also have specific criteria for the methods for evaluating educational attainment, the outcomes of low back pain, the language in which the articles were written, or the fact that the articles had to have been peer-reviewed?

Second, I needed help understanding the sentence with lines 199 - 201. Could you further improve this sentence?

Finally, there were some minor typos or missing words throughout the manuscript, which could be corrected.

Response: Your understanding is correct; we have stated in the inclusion criteria that only observational prospective studies (not clinical trials) are included, as reported in lines 134 - 135. 

Educational attainment: The specific method for assessing educational attainment was stated in lines 174 -178 “Where possible, the minimum data extracted were related to a 12-year cut-point for EA as representing the threshold between secondary (up to year 12) and post-secondary (beyond year 12) EA in most countries. Where data were not presented with adequate detail, EA was sorted into meaningful order based on the manner reported in the studies (e.g., low vs high).”

Outcome of Back Pain: Line 181 – 187: “Outcomes were limited to those broadly categorized as either pain (e.g., presence/absence of LBP or pain intensity) or disability (e.g., return to work or score on a standardized patient-reported outcome). Where studies reported “recovery” as an outcome those operationalizations were reviewed for relevance and if aligned with our purpose the verbatim definition was extracted and assigned to the most relevant outcome category (e.g., pain, disability, or both)”.

Language in which the articles were written: In line 148 under the search strategy, we state that the databases were searched without restriction to the language of publication. However, we have also now included an allusion to the English-language focus as a potential limitation to interpretation in the Limitations section of the paper: “We did not include studies published in a language other than English or without a formal English translation available, raising the risk that we have missed data from work published in non-English journals that may influence our results.”

Peered reviewed: In this study, we have attempt

---

## [Decision Letter · Decision Letter 1]

29 Jul 2024

Is educational attainment associated with the onset and outcomes of low back pain? A systematic review and meta-analysis

PONE-D-24-08660R1

Dear Dr. Lawan,

We’re pleased to inform you that your manuscript has been judged scientifically suitable for publication and will be formally accepted for publication once it meets all outstanding technical requirements.

Kind regards,

Mohammad Ali

Academic Editor

PLOS ONE

Additional Editor Comments (optional):

Reviewers' comments:

Reviewer's Responses to Questions

**Comments to the Author**

1. If the authors have adequately addressed your comments raised in a previous round of review and you feel that this manuscript is now acceptable for publication, you may indicate that here to bypass the “Comments to the Author” section, enter your conflict of interest statement in the “Confidential to Editor” section, and submit your "Accept" recommendation.

Reviewer #1: All comments have been addressed

Reviewer #2: All comments have been addressed

2. Is the manuscript technically sound, and do the data support the conclusions?

Reviewer #1: Yes

Reviewer #2: Yes

3. Has the statistical analysis been performed appropriately and rigorously? 

Reviewer #1: Yes

Reviewer #2: Yes

4. Have the authors made all data underlying the findings in their manuscript fully available?

Reviewer #1: Yes

Reviewer #2: Yes

5. Is the manuscript presented in an intelligible fashion and written in standard English?

Reviewer #1: Yes

Reviewer #2: Yes

6. Review Comments to the Author

Reviewer #1: "The corrections suggested by me were addressed, and the inquiries were answered appropriately, so the manuscript can now be published. Congratulations!"

Reviewer #2: I would like to thank the authors for responding so thoroughly to my comments. I have no further comments.

7. PLOS authors have the option to publish the peer review history of their article (what does this mean?). If published, this will include your full peer review and any attached files.

Reviewer #1: **Yes: **CARLOS AUGUSTO FERREIRA DE ANDRADE

Reviewer #2: No

---

## [Editor Report · Acceptance letter]

5 Aug 2024

PONE-D-24-08660R1 

PLOS ONE

Dear Dr. Lawan, 

I'm pleased to inform you that your manuscript has been deemed suitable for publication in PLOS ONE. Congratulations! Your manuscript is now being handed over to our production team.

Kind regards, 

on behalf of

Dr. Mohammad Ali 

Academic Editor

PLOS ONE